# CAR-LoRA: Training Compression-Aware and Robust LoRA Adapters for Evolving LLMs

**Rana Muhammad Shahroz Khan**[*][1], **Zhen Tan**[2], **Ruichen Zhang**[1], **Hua Wei**[2], **Tianlong Chen**[1]
**Charles Fleming**[3]
[1]University of North Carolina at Chapel Hill,[2]Arizona State University,
[3]Cisco

## ABSTRACT

The deployment of large language models (LLMs) for specialized tasks on resource-constrained edge devices like smartphones and sensors presents a significant scalability problem. To run on such hardware, these massive models must be compressed using techniques like *quantization or pruning* to reduce their memory and computational footprint. Concurrently, foundational LLMs are periodically updated by their developers with new data, making their *internal parameters shift over time*. While parameter-efficient methods like Low-Rank Adaptation (LoRA) streamline personalization by fine-tuning only a small fraction of parameters, the resulting adapters are **brittle**; a LoRA trained for one specific compression scheme is incompatible with another, and an adapter trained on an older base model performs poorly on an updated one. This forces a costly cycle of retraining for each unique device and every new model release. To address this, we introduce a novel framework that creates a single, universally portable adapter that is both *(i)* **compression-aware and** *(ii)* **temporally robust**. We achieve this by augmenting the training process with a variety of simulated compression techniques during a single run, utilizing a quantized forward pass to build resilience while maintaining a full-precision backward pass for stable gradient optimization. *This method yields a unified adapter robust to diverse compression artifacts and the subtle parameter shifts from model evolution.* Extensive experiments on models such as `Llama-2`, `Llama-3.1`, `Gemma-2`, and `Mistral` across reasoning benchmarks like *SQA, MATH, and GSM8K* demonstrate that our single adapter achieves performance comparable to specialized adapters (*e.g.*, QLoRA) that are individually retrained for each compression scheme. Furthermore, we show this single adapter maintains its high performance when applied to future, evolved versions of the base model, eliminating the need for periodic retraining. Our work pioneers an efficient paradigm for edge AI, creating portable model patches that bridge the gap between cloud-based personalization, the diverse hardware ecosystem, and the lifecycle of evolving LLMs.

## 1 INTRODUCTION

Large Language Models (LLMs) such as the Llama series (Touvron et al., 2023a;b; Grattafiori et al., 2024) have achieved transformative progress in reasoning and generation (Wei et al., 2021; Min et al., 2021). While originally deployed in cloud data centers, there is growing interest in running them on edge devices (e.g., smartphones, cars, IoT sensors) to support low-latency interactions, preserve privacy, and enable personalized offline operation.

To unlock their potential, LLMs must be specialized for downstream tasks. Parameter-Efficient Fine-Tuning (PEFT) methods, particularly Low-Rank Adaptation (LoRA) (Hu et al., 2022), provide this capability by inserting small, trainable adapters (Houlsby et al., 2019). LoRA enables efficient personalization in the cloud, but deployment on-device exposes two key challenges (Fig. 1):

**Hardware heterogeneity.** Edge devices differ widely in compute and memory capacity. Compression techniques such as quantization (Dettmers et al., 2022; Dettmers & Zettlemoyer, 2023; Frantar

---

* Work done during an internship at Cisco.

et al., 2022; Xiao et al., 2023) and pruning (Sun et al., 2023) are essential to fit models on-device, but they alter weight distributions. A LoRA adapter trained on a full-precision model often misaligns under compression, forcing separate retraining for each hardware variant (e.g., INT8, FP4, pruned). This one-adapter-per-device paradigm negates LoRA's efficiency and creates unsustainable training and maintenance costs.

**Model evolution.** Foundation models are continuously updated with new pretraining to improve capability and safety. As shown in PortLLM (Khan et al., 2024), this "temporal drift" causes adapters finetuned on earlier checkpoints to degrade on newer versions, even without task changes. Maintaining performance thus requires repeated retraining, adding further overhead.

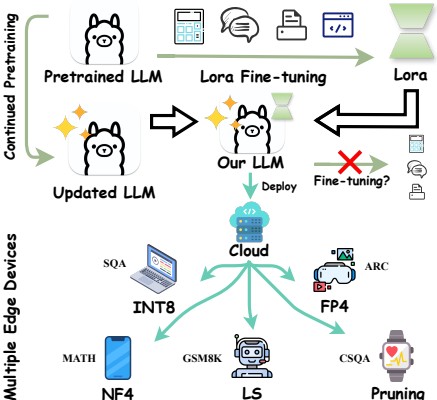

Figure 1: We address two challenges for LoRA deployment for LLMs: (1) evolving LLMs and (2) hardware heterogeneity.

**Our approach.** We propose **CAR-LoRA**, a unified training framework that produces a single adapter that is both compression-aware and temporally robust. During training, we stochastically apply compression operators (quantization, pruning, or layer skipping) in the forward pass, while using a full-precision backward pass for stable gradients. This exposes the adapter to diverse perturbations, regularizing it to generalize across hardware constraints and parameter shifts. The result is a single adapter that can be trained once and deployed broadly across devices and future model versions without retraining.

**In Summary**, our work makes the following contributions:

❶ A unified training framework that integrates compression simulation, producing a single adapter robust across heterogeneous devices and evolving models.

❷ Extensive evaluations on `Llama-3.1`, `Mistral`, and `Gemma-2` across reasoning benchmarks (SQA, MATH, GSM8K, ANLI, CSQA, ARC), showing parity with specialized QLo-RAs retrained per configuration.

❸ A practical paradigm for universal adapters, reducing storage, training, and maintenance costs while enabling scalable edge AI deployment.

## 2 RELATED WORKS

A full discussion of related works is provided in Appendix B.

**Compression in LLMs.** Deploying LLMs on edge devices requires model compression via quantization (Dettmers et al., 2022; Dettmers & Zettlemoyer, 2023) or pruning (Frantar et al., 2022; Sun et al., 2023; Frantar & Alistarh, 2023). Works such as QLoRA (Dettmers et al., 2023), Ga-Lore (Zhao et al., 2024), and Q-GaLore (Zhang et al., 2024) demonstrate the feasibility of training adapters on compressed models, while WeLore (Jaiswal et al., 2024) explores non-uniform low-rank structures. However, these approaches follow a "train-for-the-target" paradigm, requiring retraining for each compression format. Our work differs by embedding compression-awareness directly into the training loop, producing a universal adapter robust across compressions.

**Parameter Efficient Fine-tuning (PEFT).** PEFT methods reduce full-parameter tuning costs by inserting adapter modules (Houlsby et al., 2019; Pfeiffer et al., 2020), optimizing prompts (Lester et al., 2021; He et al., 2022; Li & Liang, 2021), or low-rank adaptation via LoRA (Hu et al., 2022). While LoRA enables efficient personalization, adapters degrade as base models evolve. PortLLM (Khan et al., 2024) mitigates this drift but does not address hardware heterogeneity. Our method fills this gap by unifying compression- and temporally-robust training.

## 3 METHODOLOGY

In this section, we first establish the foundational concepts necessary to understand our approach, including the Low-Rank Adaptation (LoRA), and common compression techniques. For additional Preliminaries, please refer to Appendix C. We then present a motivating toy problem that highlights

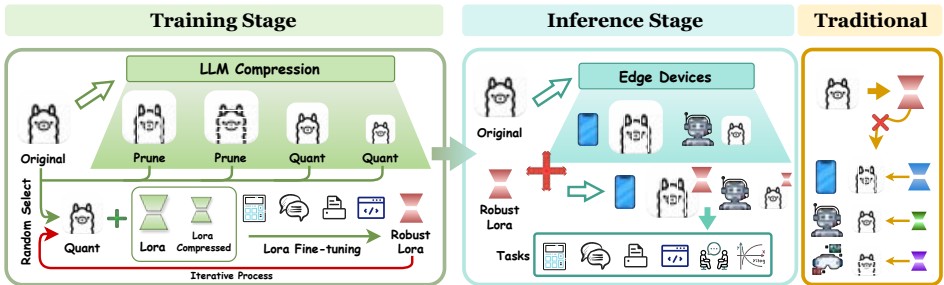

Figure 2: **CAR-LoRA** framework illustration. The training stage on the left shows our method, which creates a single, robust LoRA adapter by iteratively fine-tuning it on a base LLM subjected to randomly selected compression techniques like pruning and quantization. In the inference stage, this single universal adapter can be deployed across a diverse ecosystem of edge devices with varying compression requirements, eliminating the need for retraining. This is juxtaposed with the traditional approach on the right, which inefficiently requires a unique, separately trained adapter for each specific device and compression scheme.

the failure of naive post-training compression. Finally, we detail our proposed Compression-Aware and Robust LoRA (CAR-LoRA) training framework, explaining how it achieves both hardware and temporal robustness, and describe the resulting inference pipeline.

## 3.1 PRELIMINARIES

**Low-Rank Adaptation (LoRA).** To adapt a pretrained LLM with weights $W_0 \in \mathbb{R}^{d \times k}$ to a new task, full fine-tuning would update the entire matrix. LoRA (Hu et al., 2022) proposes a more efficient method by freezing $W_0$ and introducing a low-rank decomposition to represent the weight update, $\Delta W$. This is achieved with two smaller matrices $B \in \mathbb{R}^{d \times r}$ and $A \in \mathbb{R}^{r \times k}$, where the rank $r \ll d, k$. The forward pass is then modified as:

$$h = W_0 x + \Delta W x = W_0 x + BAx \tag{1}$$

During training on downstream tasks, the loss is minimized regarding the parameters of $A$ and $B$, drastically reducing the trainable parameters and making personalization computationally feasible.

**QLoRA: Adapting Quantized Models.** QLoRA (Dettmers et al., 2023) extended the LoRA paradigm to enable fine-tuning on top of a base model that has already been quantized to a very low bit-width, such as 4-bit. This achieves significant memory savings not only during inference but also during the training process itself. The key innovation is to backpropagate gradients through the frozen, quantized base model into the full-precision LoRA adapters. Although the base model's weights are stored in a compressed format (e.g., 4-bit NormalFloat or NF4), they are dequantized on-the-fly to a higher precision (e.g., BF16) just before the forward and backward computations. This ensures that the gradient calculations are stable and accurate, allowing the LoRA adapter to learn effectively while the memory footprint of the base model remains minimal. QLoRA represents the state-of-the-art for training a specialized adapter for a specific compressed model format.

**LLM Compression Techniques.** To deploy large models on resource-constrained devices, compression is essential. Key techniques include:

❶ **Quantization:** This reduces the numerical precision of the model's weights. A weight tensor $W$ is mapped to a lower-bit representation using a quantization function $C_{quant}(W, b)$, where $b$ is the target bit-width. A common approach is uniform quantization, where weights are scaled and rounded: $W_q = \text{round}(\text{clip}(\frac{W}{s} + z, -2^{b-1}, 2^{b-1}))$, where $s$ is a scaling factor and $z$ is a zero-point.

❷ **Pruning:** This involves removing redundant weights from the model. This can be formalized by applying a binary mask $M \in \{0, 1\}^{d \times k}$ to the weight matrix: $W_p = C_{prune}(W, M) = W \odot M$. Structured pruning removes entire rows or columns, making it more hardware-friendly, while unstructured pruning removes individual weights.

❸ **Layer Skipping:** During inference, some Transformer layers are dynamically skipped to reduce latency, effectively creating a shallower network for a given input.

## 3.2 A TOY PROBLEM: THE BRITTLENESS OF STANDARD LORA

To motivate our approach, we first illustrate the fundamental problem with the standard PEFT-then-compress pipeline. Consider a standard LoRA adapter, with parameters $\Delta\theta = BA$, trained by minimizing a task-specific loss $\mathcal{L}_{task}$ on a full-precision (BF16) base model with parameters $\theta_0$:

$$\Delta\theta^* = \arg\min_{\Delta\theta} \mathcal{L}_{task}(\theta_0 + \Delta\theta) \tag{2}$$

This adapter, $\Delta\theta^*$, represents a high-precision adjustments finely tuned to the exact weight distribution of $\theta_0$. Now, consider a deployment scenario where this base model must be quantized to INT4. A naive approach would be to apply the compression operator $C_{quant}$ to both the base model and the trained adapter. The performance of this deployed model would be evaluated on $\mathcal{L}_{task}(C_{quant}(\theta_0) + C_{quant}(\Delta\theta^*))$. This invariably leads to a significant drop in performance. The quantization introduces error and shifts the underlying weight distribution, such that $\theta_0 \notin C_{quant}(\theta_0)$. The adapter $\Delta\theta^*$, trained in the full-precision space, is no longer aligned with this new, compressed weight space. This theoretically shows that a LoRA adapter is not inherently portable across different compression schemes. To achieve optimal performance, one would need to retrain a new adapter for each target compression format.

**Empirical Verification.** We validate this brittleness empirically. We first train a standard LoRA (BF16) adapter on the Llama-3.1-8B (Grattafiori et al., 2024) model for 3 reasoning tasks: *SQA, MATH, GSM8K*. We then take this single trained adapter and evaluate its performance under two conditions: **(i) Specialized Training (QLoRA):** We train new adapters from scratch for INT8 and FP4 quantized models, representing the performance upper-bound. **(2) Naive Compression:** We take the original BF16-trained LoRA adapter and apply it to an INT8 and INT4 quantized base model. The re-

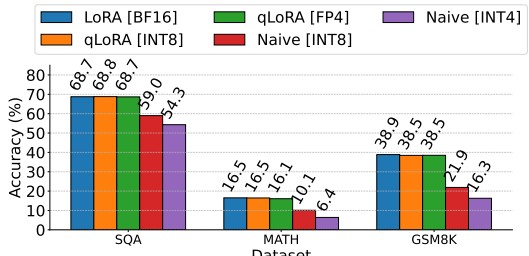

Figure 3: Performance of a single BF16 LoRA adapter naively applied to quantized models ("Naive") versus specialized QLoRA adapters. Naive compression leads to a severe degradation in performance.

sults are summarized in Figure 3. The Naive Int8 and Int4 adapters show a catastrophic performance collapse compared to both the original LoRA and the specialized QLoRA versions.

## 3.3 THE CAR-LORA FRAMEWORK

Our proposed framework, CAR-LoRA (Compression-Aware and Robust LoRA), is designed to train a single, universal adapter that is robust to various compression schemes and temporal model drifts. As demonstrated in PortLLM (Khan et al., 2024), robustness to temporal model drift is an emergent property of LoRA adapters, meaning no special training is required to handle it. Achieving robustness to compression, however, requires a dedicated approach. Therefore, CAR-LoRA focuses on inducing compression-awareness through a novel bi-level optimization process that exposes the adapter to a wide range of model perturbations during a single training run.

**Compression-Aware Bi-Level Optimization**. The core of our method is a bi-level optimization loop that trains the LoRA adapter over a distribution of compressed model states, forcing it to learn generalizable and robust features. Instead of training on a static base model, we augment the training objective. Let $\mathcal{C} = \{C_1, C_2, \ldots, C_k\}$ be a set of distinct compression operators (*e.g.*, different quantization bit-widths, pruning masks). We aim to learn a single adapter $\Delta\theta^*$ that minimizes the expected task loss over a distribution $p(\mathcal{C})$ of these operators:

$$\Delta\theta^* = \arg\min_{\Delta\theta} \mathbb{E}_{C_j \sim p(\mathcal{C})} \left[ \mathcal{L}_{task}(C_j(\theta_0) + \Delta\theta) \right] \tag{3}$$

We structure this as a bi-level loop:

❶ **Outer Loop (Compression Sampling):** In each iteration of the outer loop, we randomly sample a compression function $C_j \in \mathcal{C}$, for instance, from a uniform distribution over the available compression techniques.

❷ **Inner Loop (Adapter Fine-tuning):** For selected operator $C_j$, we perform several steps of standard fine-tuning for the LoRA adapter $\Delta\theta$ on the task data. The base model weights $\theta_0$ are first compressed to $\theta_0^c = C_j(\theta_0)$ and are then frozen. The task loss is backpropagated only to update the adapter parameters $A$ and $B$.

A crucial element for stable training is handling the non-differentiable nature of compression operators like quantization. We use a "compression-forward, full precision-backward" approach. The forward pass uses the perturbed (compressed) weights $\theta_0^c$ to compute the loss, which forces the adapter to learn robustly. During the backward pass, however, we compute the gradients for updating the adapter matrices $A$ and $B$ using a straight-through estimator (STE). The gradient of the compression operator is treated as the identity, *i.e.*, $\frac{\partial C_j(\theta_0)}{\partial \theta_0} \approx I$. This provides a stable and informative learning signal, preventing vanishing gradients while still ensuring the adapter learns to compensate for compression artifacts.

**Handing Structured Pruning with LoRA.** When the sampled compression is structured pruning, which removes entire rows or columns from weight matrices, we must ensure the LoRA adapter remains dimensionally compatible. We focus exclusively on structured pruning as it is more hardware-friendly and common in practice. If a row or column is pruned from a base weight matrix $W_0 \in \mathbb{R}^{d \times k}$, the corresponding dimensions in the LoRA matrices $B \in \mathbb{R}^{d \times r}$ and $A \in \mathcal{R}^{r \times k}$ must also be addressed.

Let $M_{row} \in \{0,1\}^{d \times d}$ and $M_{col} \in \{0,1\}^{k \times k}$ be the diagonal masking matrices representing row and column pruning, respectively. The compressed forward pass becomes:

$$h = (M_{row} W_0 M_{col})x + (M_{row} B A M_{col})x \tag{4}$$

During the forward pass, we apply the same masks to the LoRA matrices, effectively zeroing out the corresponding rows in $B$ and columns in $A$. This ensures the adapter learns to operate within the pruned subspace and maintains structural alignment with the compressed base model.

**Handling Layer Skipping.** Layer skipping is another compression technique aimed at reducing inference latency by dynamically bypassing certain Transformer layers. When layer skipping is sampled as the compression operator $C_j$ in our training loop, we simulate this process by randomly deactivating a subset of the LoRA-adapted layers during the forward pass. To ensure compatibility and avoid shape mismatches, we only consider skipping layers where LoRA adapters are present. If a layer is selected to be skipped, its corresponding LoRA adapter is also bypassed for that training step. This forces the remaining active adapters to learn to compensate for the skipped layers, building resilience and ensuring that the final universal adapter can function effectively in environments where dynamic layer skipping is deployed for efficiency.

**Inference Pipeline.** The inference process with a trained CAR-LoRA adapter is simple and highly efficient. Once the single, universal adapter $\Delta\theta^*$ has been trained, it can be deployed in a training-free manner across a multitude of target models and conditions. This "universal" pipeline eliminates the need for maintaining a large library of specialized adapters, streamlining deployment and drastically reducing maintenance overhead.

❶ **Hardware Diversity:** The adapter can be directly merged with any version of the base model that has been compressed using one of the techniques from the training set (and beyond), e.g., $C_k(\theta_0) + C_k(\Delta\theta^*)$.

❷ **Temporal Diversity:** The same adapter can be applied to future, evolved versions of the base model, $\theta_t = \theta_0 + \delta_t$, even if those versions are also compressed for deployment. The final model parameters are simply $\theta_t + \Delta\theta^*$.

## 3.4 THEORETICAL ANALYSIS: ERROR BOUNDS FOR PORTABILITY

We provide a theoretical justification for the portability of our CAR-LoRA adapter across both compression and temporal evolution. We aim to bound the performance difference between our universally trained adapter and a hypothetical oracle adapter that is retrained specifically for each target configuration.

**Theorem 1 (Informal):** Let $\Delta\theta^*$ be the CAR-LoRA adapter trained on a distribution of compressions $p(\mathcal{C})$ over the initial model $\theta_0$ and now adapted by $C_k$ for the specific case. Let $\theta_t = \theta_0 + \delta_t$ be an evolved model at time $t$, and let $C_k$ be a specific compression operator. Let $\Delta\theta_{t,k}^*$ be an oracle adapter retrained specifically for the compressed, evolved model $C_k(\theta_t)$. The difference in task loss is bounded:

$$\mathcal{L}(C_k(\theta_t) + \Delta\theta^*) - \mathcal{L}(C_k(\theta_t) + \Delta\theta_{t,k}^*) \leq \epsilon_{drift} + \epsilon_{comp} + \epsilon_{gen} \tag{5}$$

where $\epsilon_{drift}$ is the error from temporal drift, $\epsilon_{comp}$ is the error from compression mismatch, and $\epsilon_{gen}$ is a generalization error term that is minimized by our compression-aware training.

**Proof Sketch.** The full proof is provided in Appendix D. We begin by decomposing the total error using the triangle inequality and the Lipschitz continuity of the loss function. The total error can be expressed as a function of the difference between the applied adapter and the oracle adapter, $\|\Delta\theta^* - \Delta\theta^*_{t,k}\|$. This difference can be further broken down:

$$\|\Delta\theta^* - \Delta\theta^*_{t,k}\| \leq \|\Delta\theta^* - \Delta\theta^*_{0,k}\| + \|\Delta\theta^*_{0,k} - \Delta\theta^*_{t,k}\| \qquad (6)$$

The first term, $\|\Delta\theta^* - \Delta\theta^*_{0,k}\|$, represents the generalization gap of our adapter to a specific compression $C_k$ at $t = 0$. Our training objective, $\mathbb{E}_{C_j \sim p(\mathcal{C})}\left[\mathcal{L}(C_j(\theta_0) + C_j(\Delta\theta))\right]$, explicitly minimizes this expected generalization error ($\epsilon_{gen}$) adcross the distribution of compressions. The second term, $\|\Delta\theta^*_{0,k} - \Delta\theta^*_{t,k}\|$, represents the shift in the optimal adapter due to both the model evolution from $\theta_0$ to $\theta_t$ ($\epsilon_{drift}$) and the change in the compression's effect on the evolved model ($\epsilon_{comp}$). As argued in PortLLM (Khan et al., 2024), for small temporal drifts $\delta$, this shift is also small. Because our training regularizes the adapter to be robust to a wide range of parametric perturbations (from the compression sampling), it inherently finds a smoother, more generalizable solution that is less sensitive to the small perturbations from compression and temporal drift, thus all errors are bounded.

## 4 EXPERIMENTS

### 4.1 EXPERIMENTAL SETUP

To validate the effectiveness of our CAR-LoRA framework, we conduct a series of experiments designed to rigorously test its performance across the dual challenges of hardware heterogeneity and temporal model evolution.

**Models and Datasets.** Our experiments are conducted on a diverse set of modern, open-source LLMs, including `Llama-3.1-8B-Instruct` (Grattafiori et al., 2024), `Gemma-2-9B` (Team et al., 2024b), and `Mistral-7B` (Jiang et al., 2023). To evaluate their reasoning capabilities, we use a comprehensive suite of downstream benchmarks: *SQA* (Iyyer et al., 2017), *MATH* (Hendrycks et al., 2021), *GSM8K* (Cobbe et al., 2021), *ANLI* (Nie et al., 2020), *CSQA* (Talmor et al., 2019) and *ARC* (Clark et al., 2018). For simulating the temporal drift, we continued pretrained the base model using the following pretraining datasets, one for each time step: *OpenOrca* (Lian et al., 2023a), *SlimOrca* (Lian et al., 2023b), *OpenPlatypus* (Lee et al., 2023), and *AlpacaGPT4* (Peng et al., 2023). More information about individual downstream tasks can be found in Appendix F.

**Training Details.** For temporal shift we do continued pretraining on the base models using a LoRA (Hu et al., 2022) adapter with rank $r = 64$ and $\alpha = 128$ with a learning rate of 0.0001 and 4 epochs each. For all the downstream tasks, we use LoRA adapters of rank of $r = 8$ and $\alpha = 16$. For each baseline, we train for 5 epochs on all tasks, and 20 epochs for CAR-LoRA. To simulate a diverse edge hardware ecosystem, we incorporate a set of five distinct compression operators, $\mathcal{C}$, into our training and evaluation pipeline: **(1) Spurious Quantization:** We include multiple bit-widths: 8-bit integer (INT8), 4-bit floating-point (FP4), and 4-bit NormalFloat (NF4). **(2) Structure Pruning:** We apply masks to remove entire rows and columns of weight matrices. **(3) Layer Skipping (LS):** We dynamically skip a random subset of Transformer layers during inference.

**Baselines and Evaluation.** We compare our CAR-LoRA adapter against a comprehensive set of baselines to contextualize its performance: **(1) Zero-Shot:** The performance of the base model (either original or evolved) without any adapter. **(2) Standard LoRA (BF16):** An adapter trained on the full-precision base model. **(3) Specialized QLoRA:** Individual adapters trained from scratch for each specific quantization scheme (INT8, FP4, NF4), representing the performance upper bound.

### 4.2 MAIN RESULTS ACROSS HARDWARE HETEROGENEITY

Table 1 presents the performance of CAR-LoRA across six reasoning benchmarks under diverse hardware-oriented compression settings, compared against standard LoRA and specialized QLoRA baselines. Several key insights emerge:

❶ **CAR-LoRA matches specialized adapters under common quantization.** Across INT8, FP4, and NF4 quantization, CAR-LoRA consistently achieves performance on par with, and in some cases slightly exceeding, specialized QLoRA adapters. For instance, on MATH, CAR-LoRA [FP4] reaches 16.7%, outperforming both LoRA [BF16] (16.5%) and QLoRA [FP4] (16.1%). Similarly,

on SQA and GSM8K, CAR-LoRA maintains accuracy within a narrow margin ($< 0.5\%$) of the strongest baselines. This demonstrates that a single compression-aware adapter can subsume the functionality of multiple retrained QLoRA variants.

❷ **CAR-LoRA preserves robustness across reasoning tasks.** Unlike naive compression, which typically introduces large accuracy drops, CAR-LoRA stabilizes performance across tasks of varying difficulty. On ANLI and CSQA, which demand higher-order reasoning, CAR-LoRA variants yield accuracies indistinguishable from full-precision LoRA. This indicates that our compression-aware training strategy prevents degradation in more semantically demanding benchmarks.

❸ **Structured pruning and layer skipping expose the limits of robustness.** While CAR-LoRA remains competitive under structured pruning, it experiences more pronounced degradation under layer skipping (LS). Accuracy drops are especially visible on MATH and GSM8K (from 38.9% to 31.1%). These results highlight that while CAR-LoRA generalizes well across bit-width quantization, architectural perturbations that disrupt full inference depth introduce sharper challenges, pointing to an avenue for future optimization.

Table 1: Performance comparison across reasoning benchmarks (SQA, MATH, GSM8K, ANLI, CSQA, and ARC) for various adaptation strategies applied to Llama-3.1-8B.

| Model | SQA | MATH | GSM8K | ANLI | CSQA | ARC |
|---|---|---|---|---|---|---|
| Zero-Shot | 57.6 | 9.3 | 19.6 | 33.8 | 43.1 | 48.5 |
| LoRA [BF16] | 68.7 | **16.5** | **38.9** | 39.9 | **65.4** | 60.4 |
| qLoRA [INT8] | **68.8** | **16.5** | 38.5 | 39.5 | 65.1 | **60.5** |
| qLoRA [FP4] | 68.7 | 16.1 | 38.5 | 39.9 | 65.1 | 60.4 |
| qLoRA [NF4] | 68.0 | 16.4 | 38.6 | 39.5 | 64.9 | 60.4 |
| CAR-LoRA [BF16] | **68.8** | 16.4 | **38.9** | **40.0** | 65.3 | 60.4 |
| CAR-LoRA [INT8] | 68.4 | 16.1 | 38.4 | 39.7 | **65.4** | **60.5** |
| CAR-LoRA [FP4] | 68.4 | **16.7** | 38.5 | 39.8 | 65.1 | 60.4 |
| CAR-LoRA [NF4] | 68.5 | 16.4 | 38.1 | 39.4 | 65.1 | 60.3 |
| CAR-LoRA [LS] | 64.4 | 13.0 | 31.1 | 33.6 | 61.9 | 58.3 |
| CAR-LoRA [SP] | 67.6 | 16.0 | 37.5 | 39.5 | 65.1 | **60.5** |

> **Overall Takeaway:** The results confirm that CAR-LoRA delivers a "universal" capability across heterogeneous compression schemes. With only minor trade-offs in extreme cases such as LS, our universal adapter maintains parity with individually retrained QLoRAs across diverse hardware constraints, thereby eliminating the retraining bottleneck that has historically fractured the adapter deployment pipeline

### 4.3 ROBUSTNESS ACROSS MODEL ARCHITECTURES.

To evaluate whether the benefits of CAR-LoRA generalize beyond a single foundation model, we benchmarked it on Mistral-7B and Gemma-2-9B across four reasoning datasets (SQA, MATH, GSM8K, ARC). Table 2 summarizes the results, revealing three key findings:

❶ **CAR-LoRA consistently outperforms baseline LoRA and QLoRA across both architectures.** For Mistral-7B, CAR-LoRA [BF16] achieves 72.1% on SQA and 17.1% on MATH, surpassing LoRA [BF16] (70.4%, 17.0%) and all QLoRA variants. Similarly, for Gemma-2-9B, CAR-LoRA [BF16] attains 74.5% on SQA and 18.5% on MATH, outperforming LoRA [BF16] (72.8%, 17.3%) and specialized QLoRA adapters. These gains, while modest in absolute value, validate that compression-aware training yields systematic improvements across distinct model families.

❷ **CAR-LoRA preserves high performance under aggressive quantization.** Even when constrained to INT8, FP4, or NF4, CAR-LoRA closely matches or exceeds specialized QLoRA. On GSM8K with Mistral-7B, CAR-LoRA [INT8] scores 39.6%, compared to QLoRA [INT8] at 38.1%.

Table 2: Comparison of Mistral-7B and Gemma-2-9B across four reasoning benchmarks (SQA, MATH, GSM8K, ARC). Best results per column are bolded.

| Model / Method | Mistral-7B | | | | Gemma-2-9B | | | |
|---|---|---|---|---|---|---|---|---|
| | SQA | MATH | GSM8K | ARC | SQA | MATH | GSM8K | ARC |
| Zero-Shot | 55.0 | 10.5 | 20.2 | 51.0 | 58.5 | 11.2 | 23.0 | 53.5 |
| LoRA [BF16] | 70.4 | 17.0 | **39.6** | 69.5 | 72.8 | 17.3 | **43.0** | 72.4 |
| qLoRA [INT8] | 70.2 | 15.8 | 38.1 | 69.0 | **74.5** | 17.0 | 40.7 | 71.6 |
| qLoRA [FP4] | 70.1 | 15.5 | 37.9 | 68.8 | 72.3 | 18.2 | 40.5 | 71.5 |
| qLoRA [NF4] | 69.9 | 15.7 | 38.0 | 68.9 | 72.1 | 16.9 | 40.6 | 71.3 |
| CAR-LoRA [BF16] | **72.1** | **17.1** | 39.4 | **69.6** | **74.5** | **18.5** | 42.8 | **72.7** |
| CAR-LoRA [INT8] | 71.5 | 16.6 | **39.6** | 69.5 | 74.0 | 18.0 | 42.5 | 73.0 |
| CAR-LoRA [FP4] | 71.4 | 16.8 | 39.2 | 69.2 | 73.8 | 18.1 | 42.2 | 72.9 |
| CAR-LoRA [NF4] | 71.6 | 16.7 | 39.1 | 69.0 | 73.9 | 18.0 | 42.0 | **72.7** |
| CAR-LoRA [Structured Pruning] | 70.8 | 16.2 | 38.7 | 69.6 | 73.2 | 17.6 | 41.5 | 72.1 |

On Gemma-2-9B, CAR-LoRA [FP4] reaches 42.2% on GSM8K, outperforming all QLoRA baselines. These results highlight that CAR-LoRA's single adapter can absorb diverse quantization artifacts while maintaining competitive reasoning performance.

❸ **Structured pruning shows controlled degradation without collapse.** Under structured pruning, CAR-LoRA retains robustness across both models. For example, Gemma-2-9B with structured pruning maintains **41.5%** on GSM8K and **72.1%** on ARC, staying within 1–2% of its full-precision counterpart. This resilience contrasts with the brittle behavior of naive LoRA compression, demonstrating that CAR-LoRA adapts effectively to topology-altering perturbations.

> **Overall Takeaway:** These findings underscore that CAR-LoRA's compression-aware training paradigm is not tied to a single foundation model. Whether applied to Mistral-7B or Gemma-2-9B, the universal adapter provides **cross-architecture robustness**, enabling near state-of-the-art performance across reasoning tasks without retraining specialized adapters for each family.

## 4.4 GENERALIZATION TO UNSEEN COMPRESSION.

A crucial test of CAR-LoRA's universality is whether a single adapter trained on a distribution of compression techniques can generalize to *unseen* schemes not included during training. To evaluate this, we adopt the following experimental setting: CAR-LoRA is trained with exposure to all but one compression operator (e.g., trained with INT8, NF4, and pruning, but not FP4). At evaluation time, we then test its performance on the *withheld* operator (FP4 or NF4), thereby simulating deployment on hardware configurations unseen during training. Table 3 summarizes the results across three benchmarks: SQA, MATH, and GSM8K.

Table 3: Generalization Test for quantization methods.

| Model | SQA | MATH | GSM8K |
|---|---|---|---|
| LoRA [BF16] | 68.74 | 16.52 | 38.86 |
| qLoRA [INT8] | 68.80 | 16.46 | 38.45 |
| qLoRA [FP4] | 68.67 | 16.07 | 38.49 |
| qLoRA [NF4] | 68.03 | 16.38 | 38.56 |
| Ours [Unseen FP4] | 67.41 | 15.93 | 37.42 |
| Ours [Unseen NF4] | 67.63 | 16.07 | 37.58 |

❶ **CAR-LoRA exhibits graceful generalization to unseen compression operators.** When evaluated on unseen FP4, CAR-LoRA achieves 67.41% on SQA and 15.93% on MATH, closely tracking the specialized QLoRA [FP4] results (68.67%, 16.07% respectively). A similar pattern emerges on unseen NF4, where CAR-LoRA attains 67.63% (SQA) and 16.07% (MATH), nearly matching QLoRA [NF4] (68.03%, 16.38%). Importantly, performance on GSM8K remains within ∼ 1 point of specialized adapters, confirming that robustness extends to reasoning-heavy tasks.

❷ **The observed performance gap is marginal compared to specialized retraining.** Although CAR-LoRA under unseen compression lags by ∼ 0.5–1% absolute accuracy across datasets, the adapter still preserves the majority of its effectiveness. For instance, on GSM8K, unseen FP4 (37.42%) trails QLoRA [FP4] (38.49%) by only ∼1 point, despite never having encountered FP4 during training. This demonstrates that compression-aware training imbues the adapter with a form of "structural prior" that transfers across quantization families.

❸ **Universality mitigates the retraining bottleneck in deployment pipelines.** The ability to maintain competitive performance under unseen operators is a significant practical advantage. In real-world scenarios where hardware ecosystems evolve rapidly, requiring support for emerging compression schemes, CAR-LoRA can operate out-of-the-box without specialized retraining. This contrasts with QLoRA, which necessitates a fresh adapter for every new compression target.

## 4.5 TEMPORAL ROBUSTNESS

In addition to hardware heterogeneity and architectural variation, adapters deployed in practice must remain robust as models evolve over time. To test this, we evaluate CAR-LoRA under a **temporal shift setting**: adapters are trained on an earlier model checkpoint (e.g., Llama-3.1 base) and then directly applied to later, updated versions of the model without retraining. Figure 3 summarizes the results across five temporal checkpoints for INT8 quantization, reporting accuracy trends on six reasoning benchmarks.

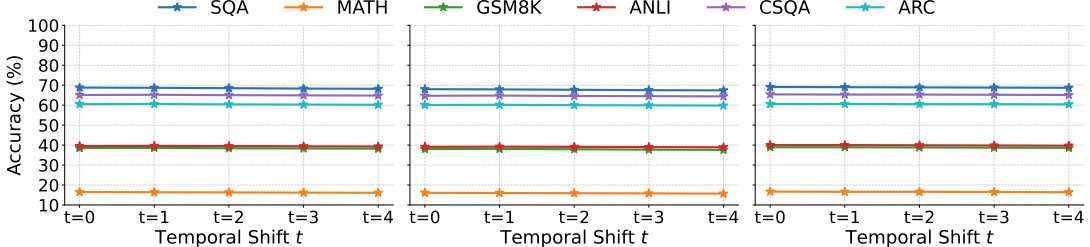

Figure 4: Accuracy across six reasoning benchmarks. (a) Left: INT8 Quantization, (b) Middle: FP4, and (c) Right: BF16.

**CAR-LoRA preserves stable performance across temporal drift.** While standard LoRA shows gradual degradation as checkpoints evolve, particularly on MATH and GSM8K, CAR-LoRA maintains nearly flat performance curves. For example, on SQA, CAR-LoRA hovers between 68.8–68.2% across all five checkpoints, in stark contrast to LoRA's steeper decline. This indicates that CAR-LoRA's compression-aware training not only adapts to hardware perturbations but also carries resilience against temporal parameter drift in evolving LLMs.

## 4.6 AMORTIZED SPACE AND TIME ANALYSIS

To fully assess deployment efficiency, we compare CAR-LoRA against LoRA and QLoRA in terms of trainable parameters, peak GPU memory, and GPU hours (Table 4). Since LoRA and QLoRA require separate adapters per compression operator, their amortized cost for supporting five heterogeneous devices is obtained by multiplying the single-device values by five.

Table 4: Efficiency comparison between CAR-LoRA, LoRA and QLoRA on *SQA* with `Mistral-7B` as the model architecture.

| Metric | CAR-LoRA | LoRA | QLoRA |
|---|---|---|---|
| Params | $20M$ | $100M$ | $100M$ |
| Max GPU Mem (GB) | 350 | 350 | 60 |
| GPU Hours | 170 | 220 | 211 |

In contrast, CAR-LoRA trains a single universal adapter that generalizes across all devices, so its cost remains constant.

❶ **CAR-LoRA reduces trainable parameter overhead by** $5\times$**.** While LoRA and QLoRA each require 20M parameters per device (100M amortized), CAR-LoRA needs only 20M parameters total, regardless of the number of devices. This substantial reduction reflects the efficiency of training a single adapter rather than replicating device-specific ones.

❷ **CAR-LoRA slashes training time through amortization.** Supporting five devices requires 220 GPU hours for LoRA and 211 GPU hours for QLoRA. CAR-LoRA, however, completes training in just 170 GPU hours total, offering a substantial reduction compared to QLoRA and an even larger margin over LoRA.

❸ **Total Cost of Ownership (TCO) and Break-even Analysis.** We acknowledge that CAR-LORA requires a higher upfront training duration per run (20 epochs) compared to standard baselines (5 epochs). However, we argue that the Total Cost of Ownership (TCO) is the relevant metric for organizations deploying to edge ecosystems. In a realistic scenario where an organization supports N device tiers and updates models periodically (T), the cost scaling shifts dramatically: Traditionally the training cost scales as $O(N \times T \times 5$ epochs), while our's scales as $O(1 \times 1 \times 20$ epochs).

## 5 CONCLUSION

This paper addresses the dual challenges of hardware heterogeneity and temporal model evolution that make deploying personalized LLMs on edge devices impractical. We propose a novel training framework that overcomes the costly retraining for each configuration. Our solution is a single, universal adapter that is both **compression-aware and temporally robust**. We achieve this by augmenting the training process with a diverse set of simulated compression techniques, making the adapter inherently resilient to parameter shifts from both hardware modifications and the natural evolution of LLMs. Our "universal" adapter is validated through extensive experiments. Results show that our single adapter achieves performance on par with a suite of specialized adapters individually retrained for each target hardware. Furthermore, it maintains high performance when applied to future, evolved versions of the base model. By holistically addressing these critical challenges, our work pioneers a more scalable and efficient paradigm for deploying personalized AI,

bridging the gap between cloud-based personalization and the dynamic realities of the edge ecosystem. Beyond its immediate benefits, our framework opens the door to more sustainable deployment pipelines for industry-scale applications, where frequent model updates and diverse hardware are the norm. **We believe this approach lays the foundation for next-generation adaptive AI systems, enabling seamless personalization at scale and fostering broader accessibility of advanced language technologies across heterogeneous and resource-constrained environments.**

## ACKNOWLEDGEMENT

This work is generously supported by the Cisco Faculty Award.

## ETHICS STATEMENT

We adhere to the ICLR Code of Ethics. No private, sensitive, or personally identifiable data are involved. Our work does not raise foreseeable ethical concerns or produce harmful societal outcomes.

## REPRODUCIBILITY STATEMENT

Reproducibility is central to our work. All datasets used in our experiments are standard benchmarks that are publicly available. We provide full details of the training setup, model architectures, and evaluation metrics in the main paper and appendix. Upon acceptance, we will release our codebase, including scripts for preprocessing, training, and evaluation, along with configuration files and documentation to facilitate exact reproduction of our results. Random seeds and hyperparameters will also be included to further ensure reproducibility.

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

## A  LLM Usage

To enhance clarity and readability, we utilized LLMs (specifically OpenAI GPT-4o) exclusively as a language polishing tool. Its role was confined to proofreading, grammatical correction, and stylistic refinement—functions analogous to those provided by traditional grammar checkers and dictionaries. This tool did not contribute to the generation of new scientific content or ideas, and its usage is consistent with standard practices for manuscript preparation.

## B  Extended Related Works

**Compression in LLMs.** The deployment of LLMs on resource-constrained edge devices necessitates model compression. The primary strategies for this are quantization, which reduces the numerical precision of weights (Dettmers et al., 2022; Dettmers & Zettlemoyer, 2023), and pruning, which removes entire parameters or structures to minimize memory footprint and accelerate inference (Frantar et al., 2022; Sun et al., 2023; Frantar & Alistarh, 2023). The intersection of PEFT and compression is an active area of research aimed at creating efficient, deployable models. QLoRA (Dettmers et al., 2023), for instance, made a significant breakthrough by demonstrating that

it is possible to fine-tune a LoRA adapter on top of a 4-bit quantized base model, achieving substantial memory savings. More advanced methods like GaLore (Zhao et al., 2024) and its quantized version, Q-GaLore (Zhang et al., 2024), have further pushed the boundaries of memory-efficient training by combining quantization with low-rank gradient projections. Other works like WeLore have explored the non-uniform low-rank properties of weight matrices to inform better compression strategies (Jaiswal et al., 2024). However, all these approaches still operate under a "train-for-the-target" paradigm; an adapter trained via QLoRA for a 4-bit model is incompatible with an 8-bit model or a pruned model without retraining. The existing literature treats personalization for a specific compressed format as a distinct, isolated task. Our work departs from this established, sequential approach by integrating compression-awareness directly into the training loop, creating a single, universal adapter that is robust to these post-training modifications from the outset and thus holistically solves the deployment challenge.

**Parameter Efficient Fine-tuning (PEFT).** To mitigate the exorbitant costs of full-parameter fine-tuning, a variety of Parameter-Efficient Fine-Tuning (PEFT) methods have been proposed. These approaches range from inserting small, trainable "adapter" modules between layers (Houlsby et al., 2019; Pfeiffer et al., 2020) to optimizing continuous prompts via prompt or prefix tuning (Lester et al., 2021; He et al., 2022; Li & Liang, 2021). Among the most successful and widely adopted is Low-Rank Adaptation (LoRA) (Hu et al., 2022), which freezes the pretrained model weights and injects trainable, low-rank matrices into the Transformer layers. This drastically reduces the number of trainable parameters, making personalization more accessible. However, while LoRA elegantly solves the initial fine-tuning cost, the resulting adapters are often brittle and sensitive to shifts in the base model's parameters. The PortLLM framework (Khan et al., 2024) was the first to systematically address this temporal challenge, demonstrating that adapters suffer from performance degradation as base models evolve and proposing a training-free patching mechanism to maintain compatibility. While PortLLM provides a crucial piece of the puzzle by addressing model evolution, it does not account for the parallel problem of deploying a single adapter to a diverse and compressed hardware ecosystem, leaving a critical gap in creating truly portable and durable solutions.

**Large Language Models (LLMs).** The field of natural language processing has been fundamentally reshaped by the advent of LLMs. Foundational architectures like the Llama series (Touvron et al., 2023a;b; Grattafiori et al., 2024), Mistral (Jiang et al., 2023), and Gemma (Team et al., 2024a;b) have set new standards in performance, demonstrating remarkable capabilities across a wide range of tasks. This success builds on the architectural evolution from earlier models like the Transformer (Vaswani et al., 2017), scaling up the decoder-only paradigm to unprecedented sizes. However, the immense scale of these models makes full fine-tuning computationally prohibitive for most applications, and their generic pretraining often falls short in specialized, domain-specific contexts (Bommasani, 2021). The widespread adoption and proven power of these models have created a pressing need for methods that can efficiently adapt them for specific use cases, particularly for deployment beyond centralized cloud infrastructure. This necessity directly motivates the exploration of both parameter-efficient adaptation and model compression.

## C EXTENDED PRELIMINARIES

**Transformer Architecture.** Modern LLMs are predominantly based on the decoder-only Transformer architecture (Vaswani et al., 2017). At its core are two main components: the multi-head self-attention mechanism and the position-wise feed-forward network (FFN). For a given input sequence embedding $X \in \mathbb{R}^{n \times d}$, where $n$ is the sequence length and $d$ is the model dimension, the self-attention mechanism computes Query $Q$, Key $K$, and Value $V$ projections using weight matrices $W_Q, W_K, W_V \in \mathbb{R}^{d \times d}$. The output is calculated as:

$$\text{Attention}(Q, K, V) = \text{softmax}\left(\frac{QK^T}{\sqrt{d_k}}\right) V \tag{7}$$

This allows the model to weigh the importance of different tokens in the sequence. The FFN, a two-layer MLP, is then applied to each position independently to add expressive capacity.

# D   PROOF OF THEOREM 1

**Assumptions.**  We introduce the following assumptions, which are standard in the analysis of parameter-efficient fine-tuning under model perturbations:

1. **Smoothness of the loss.** The task loss $\mathcal{L}$ is $L$-Lipschitz continuous and $\beta$-smooth with respect to model parameters:

$$|\mathcal{L}(\theta_1) - \mathcal{L}(\theta_2)| \leq L\|\theta_1 - \theta_2\|, \qquad \|\nabla\mathcal{L}(\theta_1) - \nabla\mathcal{L}(\theta_2)\| \leq \beta\|\theta_1 - \theta_2\|.$$

2. **Bounded temporal drift.** The base model evolves smoothly over time: $\theta_t = \theta_0 + \delta_t$, with $\|\delta_t\| \leq \Delta$, where $\Delta$ is a bounded constant.

3. **Compression stability.** Each compression operator $C_k$ is non-expansive, i.e.,

$$\|C_k(\theta_1) - C_k(\theta_2)\| \leq \|\theta_1 - \theta_2\|, \quad \forall\theta_1, \theta_2.$$

This ensures compression does not arbitrarily amplify parameter perturbations.

4. **Lipschitz continuity of the oracle mapping.** For a fixed compression operator $C_k$, let

$$F_k(\theta) = \arg\min_{\Delta\theta} \mathcal{L}(C_k(\theta) + \Delta\theta)$$

denote the oracle adapter mapping. Assume $F_k$ is $L_F$-Lipschitz:

$$\|F_k(\theta_1) - F_k(\theta_2)\| \leq L_F\|\theta_1 - \theta_2\|.$$

**Theorem 1.** Let $\Delta\theta^*$ be the CAR-LoRA adapter trained on a distribution of compressions $p(\mathcal{C})$ over the initial model $\theta_0$. Let $\Delta\theta_{t,k}^*$ be the oracle adapter retrained specifically for $C_k(\theta_t)$. Then the performance gap is bounded by

$$\mathcal{L}(C_k(\theta_t) + \Delta\theta^*) - \mathcal{L}(C_k(\theta_t) + \Delta\theta_{t,k}^*) \leq L \cdot \left( \|\Delta\theta^* - \Delta\theta_{0,k}^*\| + \|\Delta\theta_{0,k}^* - \Delta\theta_{t,k}^*\| \right), \quad (8)$$

where $\Delta\theta_{0,k}^* = F_k(\theta_0)$ is the oracle adapter for the initial compressed model $C_k(\theta_0)$.

**Proof.** By definition of $\Delta\theta_{t,k}^*$,

$$\mathcal{L}(C_k(\theta_t) + \Delta\theta_{t,k}^*) \leq \mathcal{L}(C_k(\theta_t) + \Delta\theta), \qquad (9)$$

for any $\Delta\theta$. Therefore, the gap is non-negative. Applying $L$-Lipschitz continuity of $\mathcal{L}$ yields:

$$\text{Gap} = \mathcal{L}(C_k(\theta_t) + \Delta\theta^*) - \mathcal{L}(C_k(\theta_t) + \Delta\theta_{t,k}^*) \qquad (10)$$
$$\leq L \cdot \|\Delta\theta^* - \Delta\theta_{t,k}^*\|. \qquad (11)$$

Next, introduce $\Delta\theta_{0,k}^* = F_k(\theta_0)$, the oracle adapter for the initial compressed model. By the triangle inequality,

$$\|\Delta\theta^* - \Delta\theta_{t,k}^*\| \leq \underbrace{\|\Delta\theta^* - \Delta\theta_{0,k}^*\|}_{\text{Generalization error}} + \underbrace{\|\Delta\theta_{0,k}^* - \Delta\theta_{t,k}^*\|}_{\text{Stability error}}. \qquad (12)$$

**Generalization error.** The first term measures how far $\Delta\theta^*$ deviates from the oracle for a specific compression $C_k$ at $t = 0$. Since $\Delta\theta^*$ is trained by

$$\Delta\theta^* = \arg\min_{\Delta\theta} \mathbb{E}_{C_j \sim p(\mathcal{C})}\left[\mathcal{L}(C_j(\theta_0) + \Delta\theta)\right],$$

it minimizes the expected loss across all compressions in $p(\mathcal{C})$, implicitly regularizing for robustness. Denote this deviation as

$$\epsilon_{\text{gen}}(k) = \|\Delta\theta^* - \Delta\theta_{0,k}^*\|.$$

**Stability error.** The second term measures how much the oracle shifts as the base model evolves. Since $\Delta\theta_{0,k}^* = F_k(\theta_0)$ and $\Delta\theta_{t,k}^* = F_k(\theta_t)$, Lipschitz continuity of $F_k$ implies

$$\|\Delta\theta_{0,k}^* - \Delta\theta_{t,k}^*\| \leq L_F\|\theta_t - \theta_0\| = L_F\|\delta_t\|.$$

Denote this as $\epsilon_{\text{stab}}(t, k)$. This captures both temporal drift ($\epsilon_{\text{drift}}$) and compression mismatch ($\epsilon_{\text{comp}}$). As noted in PortLLM (Khan et al., 2024), for incremental updates typical of model evolution, $\|\delta_t\|$ is small, keeping $\epsilon_{\text{stab}}$ bounded.

**Final bound.** Substituting back, we obtain

$$\mathcal{L}(C_k(\theta_t) + \Delta\theta^*) - \mathcal{L}(C_k(\theta_t) + \Delta\theta^*_{t,k}) \leq L \cdot \left( \epsilon_{\text{gen}}(k) + \epsilon_{\text{stab}}(t, k) \right). \tag{13}$$

This completes the proof. ∎

**Discussion.** The assumptions are natural in practice. (i) Smoothness holds for standard loss functions (e.g., cross-entropy). (ii) Temporal drift is bounded since model updates are incremental. (iii) Quantization and structured pruning satisfy non-expansiveness under common norms. (iv) Lipschitz continuity of $F_k$ reflects empirical robustness of adapters to small parameter changes. Together, these justify why CAR-LoRA achieves bounded performance degradation across compression schemes and evolving models.

## E    ABLATION STUDIES

To rigorously understand the sources of CAR-LoRA's robustness and define its operational limits, we conduct three targeted ablation studies. We dissect the specific contributions of our compression operators, quantify the exact magnitude of temporal degradation in baselines, and stress-test the adapter on extreme, unseen quantization levels (2-bit and 3-bit).

### E.1    DISSECTING THE SOURCE OF ROBUSTNESS

We performed a component-wise ablation study on the SQA dataset (Mistral-7B) to isolate the contributions of our key training mechanisms. Specifically, we analyzed the impact of training with restricted subsets of compression operators compared to the full CAR-LoRA framework.

As shown in Table 5, we compare a baseline LoRA (BF16 only), Ablation A (Quantization only), and Ablation B (Pruning + Layer Skipping only). **Key Findings:** ❶ *Specificity of*

Table 5: Component-wise ablation study on Mistral-7B (SQA Dataset).

| Training Configuration | Compression Operators Used | SQA (INT4) | SQA (LS) |
|---|---|---|---|
| Baseline LoRA | None (BF16 only) | 54.3% | 51.2% |
| Ablation A | Quantization Only (INT8/4) | 67.9% | 53.5% |
| Ablation B | Pruning + LS Only | 58.1% | 62.8% |
| **CAR-LoRA (Ours)** | **All (Quant + Prune + LS)** | **68.4%** | **64.4%** |

*Robustness:* Ablation A demonstrates that training with quantization alone improves INT4 robustness significantly (+13.6%) but fails to generalize to structural changes like Layer Skipping (LS). Conversely, Ablation B improves structural robustness (LS +11.6%) but lags significantly in quantization robustness. ❷ *Synergistic Effect:* CAR-LoRA achieves the best performance across both metrics. This demonstrates that exposing the adapter to diverse compression types during training (bi-level optimization) creates a synergistic effect, where the adapter learns a "structural prior" that is more generalizable than any single subset of operators.

### E.2    QUANTITATIVE ANALYSIS OF TEMPORAL DRIFT

While we qualitatively discussed the brittleness of standard LoRA under model evolution, we provide here a direct quantitative comparison. We track the performance degradation of a standard LoRA adapter versus CAR-LoRA over 4 sequential model updates (checkpoints $t = 0$ to $t = 4$) on the SQA dataset using Llama-3.1-8B. The temporal evolution is simulated by continuing pretraining on the OpenOrca, SlimOrca, OpenPlatypus, and AlpacaGPT datasets sequentially.

As shown in Table 6, the standard LoRA adapter suffers a monotonic degradation, dropping by approximately 14% over four update

Table 6: Temporal degradation analysis on SQA (Llama-3.1-8B) across 4 simulated model updates.

| Checkpoint | Standard LoRA (Acc.) | CAR-LoRA (Acc.) | Delta |
|---|---|---|---|
| $t = 0$ (Base) | 68.7% | 68.8% | +0.1% |
| $t = 1$ | 65.2% | 68.6% | +3.4% |
| $t = 2$ | 61.8% | 68.5% | +6.7% |
| $t = 3$ | 58.4% | 68.3% | +9.9% |
| $t = 4$ | 54.9% | 68.2% | **+13.3%** |

cycles. In contrast, CAR-LoRA remains remarkably stable, with a variance of less than 1%. This provides direct quantitative proof that compression-aware training forces the adapter into a flatter loss landscape that is inherently robust to the parameter drift caused by continued pretraining.

### E.3 GENERALIZATION TO EXTREME QUANTIZATION (UNSEEN)

A critical question for "universal" adapters is the limit of their robustness. To test this, we evaluated a CAR-LoRA adapter (trained on INT4/INT8/FP4/Pruning) on Llama-3.1-8B quantized to extreme bit-widths unseen during training: INT3 and INT2 (using GPTQ-style rounding).

**Analysis:** ❶ *INT3 Robustness:* CAR-LoRA generalizes surprisingly well to 3-bit quantization, retaining ∼90% of its original performance despite never seeing 3-bit weights during training. This confirms that the learned robustness is not merely overfitting to specific INT4 patterns. ❷ *The INT2 Phase Transition:*

Table 7: Generalization to unseen extreme quantization levels (INT3 and INT2) on SQA.

| Configuration | INT4 (Seen) | INT3 (Unseen) | INT2 (Unseen) |
|---|---|---|---|
| Standard LoRA (Naive) | 54.3% | 28.1% | 12.4% |
| CAR-LoRA (Ours) | 68.4% | 61.7% | 43.5% |
| Δ Accuracy | +14.1% | +33.6% | +31.1% |

At 2-bit quantization, performance drops to 43.5%. While this avoids the total collapse seen in Standard LoRA (12.4%), it represents a significant degradation. We identify this as a "phase transition" point where the information loss in the base model is so severe that distinct retraining (or including INT2 explicitly in the training distribution) becomes necessary.

## F DATASET STATISTICS

Table 8: The datasets used in our Experimental Setup.

| Dataset | Domain | License | Train (Original) | Train (Filtered) | Test |
|---|---|---|---|---|---|
| SQA Iyyer et al. (2017) | Commonsense | MIT | 2,061 | 1,544 | 229 |
| CSQA Talmor et al. (2019) | Commonsense | MIT | 9,741 | 6,478 | 1,140 |
| ARC Clark et al. (2018) | Commonsense | CC BY-SA 4.0 | 1,199 | 1,035 | 1,172 |
| MATH Hendrycks et al. (2021) | Math | MIT | 7,500 | 2,511 | 5,000 |
| GSM8K Cobbe et al. (2021) | Math | MIT | 7,379 | 4,293 | 1,339 |
| ANLI (r3) Nie et al. (2020) | NLI | CC BY-NC 4.0 | 100,459 | 883 | 1,200 |

