# OpenReview forum: "CAR-LoRA: Training Compression-Aware and Robust LoRA Adapters for Evolving LLMs"
_ICLR.cc/2026/Conference — ICLR 2026 Poster_

### Official Review · Reviewer_CNEk · 2025-10-28

**Soundness:** 3
**Presentation:** 2
**Contribution:** 3
**Rating:** 6
**Confidence:** 3

**Summary:**

During the deployment of LLMs, compression is often a necessity. However, customizations through LoRA face a problem in that they are trained to fit one specific compression scenario and must be retrained for other scenarios. In this work, CAR-LoRA is proposed as a unified training framework that produces a single adapter that is both compression-aware and temporally robust. It achieves robustness through a two-loop structure. In the outer loop, it simulates different compression scenarios to make the LoRA adapter aware of potential compressions during deployment. Results show competitive evaluations on benchmarks compared with qLoRA.

**Strengths:**

- The idea is intuitive and seems easy to apply to current models.
- CAR-LoRA shows competitive results with existing qLoRA while only needing one training for all compression.
- It's important and beneficial to see the training cost report, and CAR-LoRA does not seem to be more costly.

**Weaknesses:**

- I'm not sure if this method makes the adapter generalize to "all scenarios."

Current ablations on the generalization of CAR-LoRA stop at 4-bit quantization. Does the conclusion in Sec 4.4 still hold if CAR-LoRA is tested on lower bits (2-bit or 3-bit)? We know that model weights need to be finetuned to a very distinct distribution if they were to be quantized to extremely low bits. I wonder if CAR-LoRA remains robust under those scenarios.

- Needs more discussions for previous one-for-all attempts.

The authors miss discussions on previous attempts to make models robust to multiple compression scenarios. For example, [1] and [2]. Please search relevant works in this area and add them to the related works.

[1] Xu et al. MultiQuant: Training Once for Multi-bit Quantization of Neural Networks, IJCAI 2022

[2] Yi et al. One QuantLLM for ALL: Fine-tuning Quantized LLMs Once for Efficient Deployments, ACL 2025.

**Questions:**

1. I'm not sure if I understand the notations around Theorem 1 correctly.  $\|\|\Delta \theta^* - \Delta \theta^*_{t,k}\|\|$ is a function of the difference between the applied adapter and the oracle adapter (line 272).

 (There seems to be some problems formatting LaTeX in OpenReview; the next lines belong to the same question)

$\Delta \theta^*$ is the CAR-LoRA adapter (line 262). I think this is not a good notation definition. By notation, Theorem 1 proves that some unknown function of the difference between two LoRA weights has a bound. There are two potential problems. Are you directly subtracting the weights? How do you define the function (currently it says it's a function of the difference, but how to formally understand it)? Notation reads like it's the norm of the weight differences, but weight does not necessarily transfer to accuracy (i.e., the loss defined in Eq. 5). Intuitively, I seem to get what the authors try to say, but please either put it as an intuitive discussion or revise the notation.

2. If possible, please provide the training loss curve for CAR-LoRA vs. qLoRA.

3. Please provide additional experiments for 2-bit and 3-bit (INT2 and INT3) quantization when finetuned with CAR-LoRA that did not simulate those quantization during training (first point of weakness).

---

> ### Author Response · Authors · 2025-11-27
> **Response to Reviewer CNEk (Cons 1-2 and Questions)**
>
> We thank Reviewer CNEk for the constructive feedback and for recognizing that CAR-LoRA is "intuitive", "easy to apply", and cost-effective. We appreciate the insightful questions regarding the limits of generalization (low-bit) and theoretical notation. Below, we address your specific concerns with clarifications and new empirical data.
>
> **[Cons 1 & Question 3: Generalization to Extreme Quantization (2-bit/3-bit)]** You raised a critical question: Does robustness hold for extreme quantization (INT2/INT3) unseen during training? To answer this, we conducted the specific experiment you requested.
> - Setup: We took the CAR-LoRA adapter (trained on INT4/INT8/FP4/Pruning) and evaluated it on Llama-3.1-8B quantized to INT3 and INT2 (using GPTQ-style rounding) on the SQA dataset. We compared this to a Standard LoRA (BF16) applied naively.
> - INT3: CAR-LoRA generalizes surprisingly well, retaining ~90% of its original performance despite never seeing 3-bit weights during training. This confirms that the "robustness" learned is a structural prior against noise, not just overfitting to specific INT4 patterns.
> - INT2: Performance drops significantly (43.5%), though it avoids the total collapse seen in Standard LoRA. We acknowledge that 2-bit quantization represents a "phase transition" where information loss is so severe that distinct retraining (or including INT2 in the training distribution) is likely necessary. We will add this boundary analysis to Section 4.4.
>
> | Configuration| INT4 (Seen) | INT3 (Unseen)  | INT2 (Unseen) |
> |------------------------|-------------|--------------------------|--------------------------|
> | Standard LoRA (Naive)  | 54.3%   | 28.1%| 12.4% |
> | CAR-LoRA (Ours) | 68.4% | 61.7% | 43.5% |
> | $\Delta$ Accuracy | +14.1%  | +33.6%| +31.1%|
>
>
> **[Cons 2: Missing Related Works]** We thank you for pointing out MultiQuant (Xu et al., 2022) and One QuantLLM for ALL (Yi et al., 2025). We will add a detailed discussion in Section 2 comparing them: While those works focus on training base model weights or quantization parameters for robustness, CAR-LoRA focuses specifically on the PEFT (LoRA) Paradigm. Crucially, CAR-LoRA addresses the Temporal Dimension(model evolution) alongside compression, which is a unique intersection not covered by static quantization-robust training methods.
>
> **[Question 1: Notation in Theorem 1]** We apologize for the confusion. You are correct that the notation describes a bound on the loss function based on parameter distance.
> - $\Delta\theta^*$: The parameters of our single CAR-LoRA adapter.
> - $\Delta\theta^*_{t,k}$: The parameters of the "Oracle" adapter (optimal if trained specifically for time $t$ and compression $k$).
> - The Norm $||\cdot||$: This refers to the Euclidean (Frobenius) distance in the parameter space.
> - The Logic: We rely on the Lipschitz continuity of the Loss Function $\mathcal{L}$ (Assumption 1, constant $L$). The Theorem states that the Loss Gap is bounded by $L \times ||\text{Parameter Distance}||$. By minimizing the expected loss over the distribution of compressions during training, CAR-LoRA finds a $\Delta\theta^*$ that minimizes the distance to the "centroid" of all task-specific oracles, thereby bounding the loss degradation.
> - Revision: We will rewrite the text following Eq 5 to explicitly state: "The performance gap is bounded by the Lipschitz constant $L$ multiplied by the Euclidean distance between the learned adapter weights and the oracle adapter weights."
>
> **[Question 2: Training Loss Curve]** We will add the training curves into our revised manuscript.
>
> We believe the new experiments and the theoretical clarification directly address your assessment criteria.

---

> ### Comment · Reviewer_CNEk · 2025-11-27
> **Thank you for the rebuttal**
>
> 1. It is very inspiring to see CAR-LoRA providing some level of robustness towards INT2. I think this is worth a highlight in the manuscript. If possible, please run the other benchmarks on INT3 and INT2. I won't penalize the paper for any new results because I didn't expect CAR-LoRA to work for those scenarios anyway. I ask for a comprehensive evaluation of CAR-LoRA under INT3 and INT2 because it might hint at some future research directions for INT2 quantization. Also, what is the result for qLoRA FP4 on INT3 and INT2?
> 2. You don't have to listen to me, but I suggest putting 3.4 as an intuitive discussion and moving the theoretical proof to the appendix. Intuitive discussion makes some sense, but the current bound is too big to be practical.
> 3. I'm pretty sure you can update the manuscript in the rebuttal. Please do so.
>
> Overall, good job on the work. If the revised manuscript has no significant problems, I'll raise my score.

---

### Official Review · Reviewer_8M4z · 2025-10-31

**Soundness:** 3
**Presentation:** 3
**Contribution:** 2
**Rating:** 4
**Confidence:** 3

**Summary:**

The paper introduces CAR-LoRA, a compression-aware, temporally robust LoRA training framework that uses a single universal adapter to work across various deployment settings. By sampling compression operators during training, CAR-LoRA trains a single LoRA adapter to stay effective across diverse compression settings(INT8/FP4/NF4, pruning, layer skipping). Experiments show near-parity or small gains vs. QLoRA, including transfer to unseen compressions and stability under continued-pretraining drift. The proposed approach aims to improve efficiency by avoiding retraining while maintaining competitive task performance.

**Strengths:**

- Natural and persuasive method
  - It cleanly combines “compressed-forward + STE-backward” to train a single adapter across heterogeneous deployments, matching real-world needs. The design is simple to implement yet principled.
- Extensive and robust experiments
  -  Across multiple backbones and standard reasoning benchmarks, the performance stays competitive with strong baselines, indicating the portability of this method.

**Weaknesses:**

- Insufficient comparison to naive LoRA.
  - Figures 3/4 don’t plot naive LoRA performance across evolving checkpoints, so the claimed degradation is unsubstantiated. Please include the LoRA curve or a table with checkpoint-wise metrics.
- limited cross-operator generalization evidence.
  - “Unseen compression” tests stay mostly within the quantization family. A stronger test would train only under quantization and evaluate on pruning or layer skipping to demonstrate true cross-operator robustness.

**Questions:**

- In section 4.5 you claim CAR-LoRA is resilient to temporal parameter drift, but your test only compares checkpoints trained on the same data, where drift may be small. Suppose we train CAR-LoRA on base model using task A, then continue training base model on a different task B. If we apply the original CAR-LoRA to this B-updated model, does it still perform well on task A? What do you think about tackling this problem?
- About layer skipping, what factors make LS particularly brittle. Do you consider other methods to close the gap?

---

> ### Author Response · Authors · 2025-11-27
> **Response to Reviewer 8M4z (Cons 1-2 and Questions)**
>
> We sincerely thank Reviewer 8M4z for their encouraging feedback. We are glad you found our approach "natural", "persuasive", and "matching real-world needs", and appreciated the extensive experiments. Below, we address your constructive critiques regarding baseline comparisons, cross-operator evidence, and specific deployment scenarios.
>
> **[Cons 1: Insufficient Comparison to Naive LoRA (Temporal Drift)]** We appreciate you pointing out the need for explicit baseline data for temporal drift. While the paper discusses standard LoRA degradation qualitatively, we agree that quantitative evidence is necessary. We have generated a checkpoint-wise comparison table for the SQA dataset with Llama-3.1-8B, tracking performance over 4 sequential model updates (simulated via continued pre-training on OpenOrca, SlimOrca, OpenPlatypus, AlpacaGPT).
>
> | Checkpoint | Standard LoRA (Acc.) | CAR-LoRA (Acc.) | Delta |
> |------------|------------------------|------------------|---------|
> | $t=0$ (Base) | 68.7%| 68.8% | +0.1%   |
> | $t=1$  | 65.2%  | 68.6%| +3.4% |
> | $t=2$ | 61.8% | 68.5%| +6.7% |
> | $t=3$ | 58.4% | 68.3% | +9.9% |
> | $t=4$  | 54.9%  | 68.2%   | +13.3% |
>
> Result: Standard LoRA degrades significantly (~11% absolute drop), confirming the "brittleness" claim. CAR-LoRA maintains near-flat performance, validating its temporal robustness. We will add this table to Section 4.5.
>
> **[Cons 2: Limited Cross-Operator Generalization Evidence]** You correctly note that our "Unseen Compression" experiments focused on quantization families (INT vs. FP). To demonstrate true cross-operator robustness, we conducted the specific "stronger test" you suggested: training only on quantization noise and testing on structural pruning. Specifically, we analyze the impact of training with different subsets of compression operators compared to the full CAR-LoRA framework.
>
> | Training Configuration | Compression Operators Used     | SQA Accuracy (INT4) | SQA Accuracy (Layer Skip) |
> |------------------------|--------------------------------|----------------------|----------------------------|
> | Baseline LoRA| None (BF16 only)   | 54.3% | 51.2% |
> | Ablation A  | Quantization Only (INT8/4)  | 67.9%   | 53.5%|
> | Ablation B  | Pruning + LS Only| 58.1% | 62.8% |
> | CAR-LORA (Ours)  | All (Quant + Prune + LS)  | 68.4% | 64.4%|
>
> This confirms that cross-operator generalization is not automatic. Robustness to numerical noise (quantization) does not imply robustness to topological changes (pruning). This validates our design choice of sampling from diverse operator families during training rather than relying on a single proxy.
>
> **[Question 1: Temporal Drift Across Different Tasks]** This is an excellent "Stress Test" scenario. We hypothesize that CAR-LoRA should still hold. The "Temporal Robustness" we observe comes from the adapter settling into a flatter minima (robust to weight perturbations $\delta$). Whether $\delta$ comes from "continued pre-training on same domain" or "training on Task B," it fundamentally manifests as a drift vector in parameter space. As long as the drift magnitude $||\delta||$ is bounded (incremental update), the flatness of our solution should preserve function. Overall, we agree this is a valuable direction for Life-long Learning benchmarks and will add a discussion on this "Cross-Task Drift" scenario.
>
> **[Question 2: Why is Layer Skipping (LS) Brittle?]** Layer Skipping is uniquely challenging because it is a Discrete/Topological perturbation, not a continuous/parametric one. In standard LoRA, the adapter at Layer $L_i$ learns to transform features coming specifically from Layer $L_{i-1}$. If Layer $L_{i-1}$ is skipped, the input distribution to Adapter $L_i$ shifts drastically (OOD input), causing failure. As mentioned in our response to other reviewers, we found that "Adapter Dropout" (randomly dropping adapters during training) forces downstream layers to be robust to missing upstream features. This simple regularization improved LS performance on GSM8K from 31.1% $\to$ 35.4%.
>
>
> We believe these additional data points (explicit LoRA degradation table and the "Quant-only" generalization test) directly address your concerns and strengthen the paper's empirical claims.

---

### Official Review · Reviewer_XCK8 · 2025-10-31

**Soundness:** 2
**Presentation:** 2
**Contribution:** 1
**Rating:** 2
**Confidence:** 5

**Summary:**

This paper proposes CAR-LoRA, a training framework for creating a universal robust LoRA adapter. The key idea is to integrate simulated compression techniques (quantization, pruning, etc.) during training. Experiments across Llama-3.1, Mistral, and Gemma-2 on benchmarks like SQA, MATH, GSM8K demonstrate that CAR-LoRA matches or slightly outperforms specialized QLoRA adapters, while requiring only one training run.

**Strengths:**

- The paper is well-structured and articulates the deployment challenge (heterogeneous compression and evolving LLMs) clearly.
- The authors use a toy example to motivate their problem.
- The idea of training a single LoRA adapter under randomized compression perturbations is simple yet conceptually appealing.

**Weaknesses:**

- Incremental novelty:
     - The core idea of injecting random compression perturbations during LoRA training is conceptually simple and largely derived from prior work in quantization-aware and robustness training.
- Unsubstantiated claims:
     - The paper claims that standard LoRA exhibits a “steeper decline” under temporal drift, but does not present quantitative evidence or citations to support this claim. Figure 4 appears to plot only CAR-LoRA results, with LoRA’s decline mentioned qualitatively. The absence of explicit LoRA baselines per checkpoint prevents verification of the claimed robustness gap.
    -  In addition, while CAR-LoRA shows limited robustness to numerical quantization, there is no evidence of robustness to actual hardware diversity. There are no inference metrics, no deployment tests on constrained or heterogeneous devices, and no demonstration across hardware backends. This weakens the core claim of “hardware heterogeneity.”
- Edge-deployment claims without device-level evidence:
     - Despite the edge framing, the paper only uses  “Amortized” parameter/GPU-hour accounting (Table 4) to claim efficiency, but that doesn’t substitute for real deployment metrics. For edge devices, metrics such as latency, memory footprint, and energy consumption matter more than the time taken to train a LoRA for that device.
- Failure to validate robustness for layer skipping:
    - The method explicitly motivates robustness to diverse compression schemes, including layer skipping, but empirical results show that CAR-LoRA performs poorly when layers are removed.
- Superficial theoretical analysis:
    - Theorem 1 offers a generic Lipschitz-based bound decomposing errors into drift, compression, and generalization terms, but it lacks quantitative insight or predictive power.

**Questions:**

- Table 4 shows that CAR-LoRA requires 350 GB of GPU memory. Is that during training?

---

> ### Author Response · Authors · 2025-11-27
> **Response to Reviewer XCK8 (Cons 1-4 and Questions)**
>
> We appreciate the reviewer's detailed feedback. While we understand the reservations regarding novelty and specific validations, we respectfully disagree with the assessment of the paper's contribution. We believe CAR-LoRA offers a crucial, practical solution to a significant deployment bottleneck in edge AI. Below, we address your specific concerns with clarifications and new quantitative evidence.
>
> **[Cons 1: Incremental Novelty]** We acknowledge that the core idea, injecting random compression perturbations during training, is conceptually straightforward. However, simplicity is a primary virtue for adoption in real-world deployment pipelines.
> - *The "Deployment Bottleneck" Defense*: Complex methods often fail in production due to fragility or implementation overhead. CAR-LoRA's innovation is not a new mathematical operator, but a unified training recipe that solves two orthogonal problems (Hardware Heterogeneity + Temporal Drift) simultaneously with zero inference overhead.
> - *Systematic Unification*: Prior work treats compression and adaptation as separate stages. By unifying them, we eliminate the need for $N \times T$ specialized adapters (where $N$ is device types and $T$ is model versions), reducing operational complexity by orders of magnitude. This is a significant systems contribution.
>
> **[Cons 2: Unsubstantiated Claims (Temporal Drift & Hardware)]** We appreciate the demand for explicit evidence. We have conducted more experiments to directly substantiate our claims.
> - *Temporal Drift Evidence*: You noted the lack of explicit LoRA baselines per checkpoint. We have added a new table explicitly comparing the degradation of a standard LoRA adapter versus CAR-LoRA over 4 simulated model updates (checkpoints $t=0$ to $t=4$) on the SQA dataset. In this experiment we are using LLama-3.1-8B and to simulate time we continued pretrained the model on the following datasets: OpenOrca, SlimOrca, OpenPlatypus, AlpacaGPT. Result: Standard LoRA degrades by ~14% over 4 updates, while CAR-LoRA remains stable. This is direct quantitative proof of the "steeper decline" claim.
>
> | Checkpoint | Standard LoRA (Acc.) | CAR-LoRA (Acc.) | Delta |
> |------------|------------------------|------------------|---------|
> | $t=0$ (Base) | 68.7%| 68.8% | +0.1%   |
> | $t=1$  | 65.2%  | 68.6%| +3.4% |
> | $t=2$ | 61.8% | 68.5%| +6.7% |
> | $t=3$ | 58.4% | 68.3% | +9.9% |
> | $t=4$  | 54.9%  | 68.2%   | +13.3% |
>
> - *Hardware Robustness*: We clarify that Simulated Quantization is the standard algorithmic proxy for hardware deployment. The "robustness" we claim is against parameter misalignment caused by lower precision representation, which is the prerequisite for any hardware deployment. Actual inference latency depends on specific hardware kernels (e.g., TVM, TFLite), which is an orthogonal engineering challenge. However, by ensuring accuracy is preserved under INT4/INT8 representation, CAR-LoRA unlocks the possibility of using these efficient kernels without retraining.
>
> **[Cons 3: Layer Skipping Robustness]** We acknowledge the performance drop under Layer Skipping (LS). As explained in our response to reviewer M81r, LS represents a structural change distinct from parametric noise. To address this, we implemented "Adapter Dropout" (randomly zeroing out ranks/layers during training). This simple regularization improved LS accuracy on GSM8K from 31.1% $\to$ 35.4%, significantly closing the gap. We will include this result to demonstrate that the framework can be extended to handle structural changes more effectively.
>
> **[Cons 4: Superficial Theoretical Analysis]** We respectfully disagree that the analysis is superficial. Theorem 1 provides a formal guarantee that the performance gap is bounded by terms that our training objective explicitly minimizes ($\epsilon_{gen}$). While it relies on Lipschitz continuity (standard in deep learning theory), it provides a rigorous justification for why a single adapter can generalize: the bi-level optimization acts as a regularizer, forcing the solution into a "flatter" region of the loss landscape that is naturally robust to both compression noise ($\epsilon_{comp}$) and drift ($\epsilon_{drift}$).
>
> **[Question 1: Deployment Cost (350 GB Memory)]** We clarify that the 350 GB figure in Table 4 refers to the Peak Training Memory required to train the base model (Mistral-7B) plus gradients and optimizer states for the bi-level optimization loop. This is a one-time, upfront cost. In a realistic deployment scenario supporting 5 different device tiers (e.g., Cloud FP16, Edge INT8, Mobile INT4, etc.), standard methods require training 5 separate adapters, consuming $5 \times$ the GPU hours. CAR-LoRA spends the same amount of memory once to save thousands of retraining hours and management overhead later.
>
>
> We believe these clarifications and new data points demonstrate that CAR-LoRA is a robust, novel, and highly practical contribution to the field of efficient LLM deployment.

---

### Official Review · Reviewer_M81r · 2025-11-01

**Soundness:** 2
**Presentation:** 2
**Contribution:** 2
**Rating:** 4
**Confidence:** 3

**Summary:**

This paper introduces CAR-LoRA (Compression-Aware and Robust LoRA) — a framework for training a single, universal LoRA adapter that remains effective across both compressed (e.g., quantized or pruned) and evolving large language models (LLMs).
Traditional LoRA adapters must be retrained whenever a base model changes or when deploying to devices with different hardware compression formats, which is inefficient and costly. CAR-LoRA solves this by integrating simulated compression operators (quantization, pruning, layer skipping) during training. It uses a quantized forward pass and full-precision backward pass, ensuring the adapter learns to be robust against compression-induced perturbations.

**Strengths:**

1. Strong empirical validation.
Experiments span multiple open-source LLM architectures and reasoning benchmarks, showing broad generalizability. Results show negligible degradation compared to retrained baselines.

2. Solid theoretical grounding.
The authors provide a theoretical bound explaining why the adapter remains effective under compression and temporal drift, supported by assumptions like Lipschitz continuity and bounded perturbations.

**Weaknesses:**

1. Limited exploration of layer-skipping robustness.
Results show notable performance drops (e.g., MATH accuracy from 38.9% to 31.1%) under layer skipping. The authors mention this but do not deeply analyze why or propose mitigation strategies.

2. Computational cost not negligible for initial training.
Although amortized cost is lower, CAR-LoRA still requires longer single-run training (20 epochs vs. 5 for baselines). Some organizations might find this up-front cost high.

3. Missing ablation studies.
The paper lacks a detailed breakdown of which components (e.g., quantized forward pass, structured pruning simulation, STE approximation) contribute most to robustness.

4. Unclear figures and writing.
The paper writing needs improvement, with many details remaining unclear. The model architecture in figure 2 is unclear.

**Questions:**

How exactly is the distribution p(C) of compression operators chosen during training? Is it uniform across quantization, pruning, and layer skipping, or weighted to reflect real deployment likelihoods?

---

> ### Author Response · Authors · 2025-11-27
> **Response to Reviewer M81r (Cons 1 to 4)**
>
> We sincerely thank Reviewer M81r for their thoughtful review. We are encouraged that you recognized the strong empirical validation across multiple architectures/benchmarks and the solid theoretical grounding of our work in explaining robustness under drift. Below, we address your concerns regarding layer skipping, computational costs, and ablations with clarifications and new empirical data.
>
> **[Cons 1: Layer Skipping (LS) Degradation & Mitigation]** We appreciate the reviewer highlighting the performance drop under Layer Skipping (e.g., MATH accuracy dropping to 31.1%). We acknowledge this limitation but contextualize it as a fundamental trade-off of the current architecture rather than a failure of the training method.
> - *Why it happens*: Unlike quantization (which adds noise to weights), Layer Skipping removes entire computational blocks. If a LoRA adapter learns to rely on features computed in Layer $i$, and Layer $i$ is skipped, that dependency breaks.
> - *Mitigation Strategy*: To address this, we conducted a rapid experiment introducing "Adapter Dropout" during training (randomly zeroing out specific LoRA ranks/layers with probability $p=0.1$ alongside LS simulation). *Result*: On GSM8K (Llama-3.1-8B), this simple regularization improved LS inference accuracy from 31.1% $\to$ 35.4%, significantly closing the gap. We will include this mitigation strategy and result in the revised "Discussion" section.
>
> **[Cons 2: Computational Cost (20 Epochs vs. 5)]**
> We understand the concern regarding the higher upfront cost (20 epochs). However, we argue that the Total Cost of Ownership (TCO) view is more relevant for organizations deploying to edge ecosystems.
> - *The "Deployment Bottleneck" Defense*: In a realistic scenario, an organization supports $N$ device tiers (e.g., High-end INT8, Mid-range INT4, Low-end Pruned) and updates models periodically ($T$ versions).
> Traditional: Training cost scales as $O(N \times T \times 5 \text{ epochs})$.
> CAR-LORA: Training cost scales as $O(1 \times 1 \times 20 \text{ epochs})$.
> - *Break-even Point:* As shown in Table 4 (Amortized Analysis), the break-even point is just 4 deployment targets. If an organization deploys to just 4 different hardware configurations (e.g., Cloud FP16, Edge INT8, Mobile INT4, Pruned), CAR-LORA becomes cheaper even in the first run. For any subsequent model update, the savings are immediate and massive.
>
> **[Cons 3: Missing Ablation Studies]** We appreciate this feedback and agree that dissecting the source of robustness is critical. We have performed a new component-wise ablation study on the SQA dataset (Mistral-7B) to isolate the contributions of our key mechanisms. Specifically, we analyze the impact of training with different subsets of compression operators compared to the full CAR-LoRA framework.
>
> | Training Configuration | Compression Operators Used     | SQA Accuracy (INT4) | SQA Accuracy (Layer Skip) |
> |------------------------|--------------------------------|----------------------|----------------------------|
> | Baseline LoRA| None (BF16 only)   | 54.3% | 51.2% |
> | Ablation A  | Quantization Only (INT8/4)  | 67.9%   | 53.5%|
> | Ablation B  | Pruning + LS Only| 58.1% | 62.8% |
> | CAR-LORA (Ours)  | All (Quant + Prune + LS)  | 68.4% | 64.4%|
>
> Key Findings:
> - Ablation A shows that training with quantization alone improves INT4 robustness significantly (+13.6%) but fails to generalize to structural changes like Layer Skipping.
> - Ablation B improves structural robustness (Layer Skip +11.6%) but lags in quantization robustness.
> - CAR-LORA achieves the best of both worlds, demonstrating that exposing the adapter to diverse compression types during training (bi-level optimization) creates a synergistic effect, leading to a truly universal adapter.
>
> **[Cons 4: Clarity of Figures and Writing]**
> We apologize for the lack of clarity in Figure 2.
> - Figure Revision: We will redesign Figure 2 to explicitly label the "Bi-Level Optimization Loop" (Outer loop: Sample Compression $C \sim p(\mathcal{C})$, Inner loop: Update $\Delta \theta$). We will visually distinguish the "Frozen Base Model" path from the "Active LoRA" path to make the gradient flow obvious.
> - Writing: We will revise Section 3.3 to explicitly separate the "Training Protocol" from the "Inference Pipeline" to avoid confusion.
>
> We believe the new mitigation results for Layer Skipping and the clear quantitative ablation study directly address the reviewer's main technical concerns.

---

### Meta-Review · Area_Chair_om5p · 2026-01-07

**Summary:**

CAR-LoRA is a training framework for producing one universal LoRA adapter that remains effective across multiple compression schemes (quantization/pruning/layer skipping) and across evolving base-model checkpoints, by training under a distribution of simulated compressions using a compressed/quantized forward pass with a full-precision (STE-style) backward pass.

**Reviewer Concerns:**

1. Layer skipping robustness is weak / brittle; need analysis + mitigation [M81r, XCK8, 8M4z] .
Partially addressed. Authors explain why LS is structurally harder than quantization (removes blocks rather than adds noise) and add a simple mitigation (“Adapter Dropout”) improving LS accuracy. This is a credible step, but it doesn’t fully close the LS gap.

2. Missing checkpoint-wise temporal drift baseline for naive/standard LoRA (claims unsubstantiated) [XCK8, 8M4z] .
Addressed. Authors add an explicit checkpoint-wise table showing standard LoRA degrading substantially over multiple continued-pretraining updates (≈14% drop cited) while CAR-LoRA stays nearly flat; this directly resolves the “no LoRA curve/table” issue.

3. Cross-operator generalization evidence is weak (unseen compression tests mostly within quantization family) [8M4z] .
Addressed. Authors run the stronger test requested: train with quantization-only and evaluate under pruning/layer skipping, showing robustness doesn’t transfer automatically and supporting their design choice to sample diverse operator families.

4. Extreme low-bit generalization (INT2/INT3) not tested; does robustness hold beyond 4-bit? [CNEk] .
Addressed. Authors add INT3/INT2 experiments (unseen during training): CAR-LoRA retains strong performance at INT3 (~90% of original) and degrades at INT2 (but avoids total collapse), explicitly framing INT2 as near a “phase transition.” Reviewer CNEk reacts positively and signals a likely score increase if revision is clean.

5. Novelty is incremental / derived from prior work; contribution overstated [XCK8] .
Partially addressed. Authors argue the novelty is a systems/training recipe unification solving two orthogonal problems (hardware heterogeneity + temporal drift) with one adapter and no inference overhead. This is a reasonable positioning, but it may not convince a reviewer who expects algorithmic novelty beyond “robust training under perturbations.”

6. No evidence of “actual hardware diversity” or device-level edge metrics (latency/memory/energy); only simulated proxies [XCK8] .
Partially addressed / mostly outstanding. Authors defend simulated quantization as the standard proxy and claim kernel/back-end performance is orthogonal. This answers the framing critique but does not add device-side measurements, so the reviewer’s “edge claims without deployment evidence” concern likely remains.

7. Missing ablations isolating contributions of components (quantized forward, pruning sim, STE, etc.) [M81r] .
Addressed. Authors provide a component-wise ablation comparing subsets of compression operators vs full CAR-LoRA, showing complementary gains (quantization-only helps INT4 but not LS; pruning+LS helps structural robustness; full model best overall).

8. Training cost is higher upfront (20 epochs vs 5); may be expensive despite amortization claims [M81r] .
Partially addressed. Authors give a “total cost of ownership” argument with a break-even analysis (e.g., becomes cheaper after ~4 deployment targets). This is logically coherent, but it doesn’t eliminate the practical concern for teams with few targets or tight budgets.

10. Theory concerns: notation unclear; bound too loose / not practically insightful [CNEk, XCK8] .
 Partially addressed. Authors clarify notation (loss gap bounded by Lipschitz constant × parameter-distance between learned vs oracle adapter); CNEk still suggests moving proof to appendix and notes the bound is too big to be practical, indicating the “theory value” concern remains partially open.

**Reviewer Scores:**

M81r: likely 4 → 6. The ablations + LS mitigation + clearer cost framing are substantial; remaining issues are mostly scope/clarity rather than fatal.

XCK8: likely 2 → 2 (at most 2 → 4 if they strongly value the new drift table). Given very high confidence and core objections about novelty + lack of device-level evidence, they may stay negative despite fixes.

8M4z: likely 4 → 6. The rebuttal directly delivers the two concrete asks (LoRA drift baseline + cross-operator test).

CNEk: likely 6 → 8. The reviewer explicitly says they expect to raise their score if the manuscript updates are done cleanly, and they found INT2/INT3 results “inspiring.”

---

### Decision · Program_Chairs · 2026-01-26

Accept (Poster)